A new species of Metopocetus (Cetacea, Mysticeti, Cetotheriidae) from the Late Miocene of the Netherlands

Marx Felix Georg 1 felix.marx@otago.ac.nz
Bosselaers Mark E.J. 2 3
Louwye Stephen 4
1 Department of Geology and Palaeontology, National Museum of Nature and Science , Tsukuba , Japan
2 Directorate of Earth and History of Life, Royal Belgian Institute of Natural Sciences , Brussels , Belgium
3 Marine Vertebrates, Koninklijk Zeeuwsch Genootschap der Wetenschappen , Middelburg , The Netherlands
4 Department Geology/Research Unit Palaeontology, Ghent University , Ghent , Belgium
Pyenson Nicholas
Electronic publication date: 2016 Jan 28
Publication date: 2016
Volume: 4
Electronic Location ID: e1572
Received 2015 Aug 27; Accepted 2015 Dec 16
Copyright: ©2016 Marx et al.
Copyright year: 2016
Copyright holder: Marx et al.
License: This is an open access article distributed under the terms of the Creative Commons Attribution License, which permits unrestricted use, distribution, reproduction and adaptation in any medium and for any purpose provided that it is properly attributed. For attribution, the original author(s), title, publication source (PeerJ) and either DOI or URL of the article must be cited.
License URL: https://creativecommons.org/licenses/by/4.0/

Keywords: Mysticeti, Baleen whales, Cetotheriidae, Metopocetus, Phylogenetics, Paroccipital concavity, Late Miocene, Primary dorsal infraorbital foramen

Funding: Japan Society for the Promotion of Science Postdoctoral Fellowship for Foreign Researchers FGM was supported by a Japan Society for the Promotion of Science Postdoctoral Fellowship for Foreign Researchers. The funders had no role in study design, data collection and analysis, decision to publish, or preparation of the manuscript.

==============================
The family Cetotheriidae has played a major role in recent discussions of baleen whale phylogenetics. Within this group, the enigmatic, monotypic Metopocetus durinasus has been interpreted as transitional between herpetocetines and other members of the family, but so far has been restricted to a single, fragmentary cranium of uncertain provenance and age. Here, we expand the genus and shed new light on its phylogenetic affinities and functional morphology by describing Metopocetus hunteri sp. nov. from the Late Miocene of the Netherlands. Unlike the holotype of M. durinasus, the material described here is confidently dated and preserves both the tympanic bulla and additional details of the basicranium. M. hunteri closely resembles M. durinasus, differing primarily in its somewhat less distally expanded compound posterior process of the tympanoperiotic. Both species are characterised by the development of an unusually large fossa on the ventral surface of the paroccipital process, which extends anteriorly on to the compound posterior process and completely floors the facial sulcus. In life, this enlarged fossa may have housed the posterior sinus and/or the articulation of the stylohyal. Like other cetotheriids, Metopocetus also bears a well-developed, posteriorly-pointing dorsal infraorbital foramen near the base of the ascending process of the maxilla, the precise function of which remains unclear.

Introduction

The Cetotheriidae play a crucial role in the evolution of baleen whales (Mysticeti). Long degraded to the state of a wastebasket taxon comprising nearly all fossil toothless mysticetes, the past decade saw the family restored to its original definition—Cetotherium Brandt, 1843 and relatives—within a phylogenetic context (Bouetel & De Muizon, 2006; Brandt, 1873; Steeman, 2007; Whitmore & Barnes, 2008). The importance of this prominent family lies not only in its rather disparate morphology, which is clearly distinct from that of all living species and persisted as late as the Pleistocene (Boessenecker, 2013), but also the still controversial idea that it may have given rise to the most enigmatic of the extant mysticetes, the pygmy right whale Caperea marginata Gray, 1846 (Fordyce & Marx, 2013; Marx et al., 2013; Marx & Fordyce, 2015). The phylogenetic position of the family relative to crown mysticetes remains a matter of debate, as does its exact composition and the interrelationships of the included species (Bisconti, 2015; Bouetel & de Muizon, 2006; Deméré et al., 2008; El Adli, Deméré & Boessenecker, 2014; Gol’din & Startsev, 2014; Gol’din, Startsev & Krakhmalnaya 2014; Kimura & Hasegawa, 2010; Marx & Fordyce, 2015; Steeman, 2007).

There is wide agreement on the existence of at least one subfamily, Herpetocetinae, within Cetotheriidae, comprising at least the closely related genera Herpetocetus Van Beneden, 1872 and Nannocetus Kellogg, 1929 (Whitmore & Barnes, 2008). The remaining cetotheriids are often partially or entirely lumped into the subfamily Cetotheriinae, although the definition of this grouping tends to vary across analyses (Bisconti, 2015; El Adli, Deméré & Boessenecker, 2014; Gol’din & Startsev, 2014; Marx & Fordyce, 2015; Tarasenko & Lopatin, 2012). Within this context, the genus Metopocetus Cope, 1896 has been interpreted as a potentially intermediate form linking herpetocetines and cetotheriines (Whitmore & Barnes, 2008); however, so far this taxon has had an unstable phylogenetic history (El Adli, Deméré & Boessenecker, 2014; Gol’din & Startsev, 2014; Marx & Fordyce, 2015; Steeman, 2007).

At least in part, the uncertainty surrounding Metopocetus likely reflects the incomplete nature of the available material: to date, the genus has remained restricted to its type species, M. durinasus Cope, 1896, which in turn is based on just a single, fragmentary cranium (USNM 8518) missing the rostrum, tympanic bulla and much of the basicranium (Cope, 1896; Kellogg, 1968; Whitmore & Barnes, 2008). The affinities of the only other putative occurrence of Metopocetus, “M.” vandelli (Van Beneden, 1871) from the Late Miocene of Portugal (Kellogg, 1941), are doubtful (El Adli, Deméré & Boessenecker, 2014; Gol’din & Startsev, 2014; Whitmore & Barnes, 2008). Compounding these issues further are the lack of clear stratigraphic and provenance data for USNM 8518, which may have been derived from either Langhian or Tortonian deposits (Case, 1904; Kellogg, 1931; Kellogg, 1968).

Here, we describe a new species of Metopocetus from the Late Miocene of north-western Europe (the Netherlands), the first material clearly representing this genus besides M. durinasus, and its first occurrence outside North America (Fig. 1). Unlike USNM 8518, the specimen described here is confidently dated and preserves both the tympanic bulla and additional details of the basicranium, thus providing new insights into cetothere phylogeny and functional morphology.

Figure 1 Type locality of Metopocetus hunteri.

Drawing of cetotheriid by Carl Buell.

Material and Methods

Collection, preparation and phylogenetic analysis

The specimen was collected in 1987 by O. Stolzenbach and mechanically prepared by K. Post and one of the authors (MB). Morphological terminology follows Mead & Fordyce (2009), unless indicated. For the figures, photographs of the specimen were digitally stacked in Photoshop CS6. To determine the phylogenetic position of our new material, we added the specimen to the recently published matrix of Marx & Fordyce (2015: Fig. 2). Further, we also included “Metopocetus” vandelli (holotype MUHNAC A1) and the morphologically similar “Aulocetus” latusKellogg, 1941 (holotype MUHNAC A2) to determine their placement relative to Metopocetus proper. Both of these taxa are known only from Adiça (Lower Tagus Basin, Portugal) and were recovered from Late Miocene strata correlative with Cotter’s lithostratigraphic zone VIIb, dated to ca 9.5–8.5 Ma (Antunes et al., 2000; Estevens & Antunes, 2004; Kellogg, 1941; Pais, Legoinha & Estevens, 2008).

Besides these additions, we retained all of the previous taxa and codings, with two exceptions: in the previous analysis “Cetotherium” megalophysum Cope, 1895, was coded as having the posterior end of the ascending processes of the maxillae contact each other in dorsal view (char 69:2), and consequently as “NA” for character 68, “Triangular wedge of frontal separating ascending process of maxilla from nasal or premaxilla”. Further observations have revealed these observations to be inaccurate, and we here correct them to states 68:0 (triangular wedge of frontal absent) and 69:1 (ascending processes of maxillae converging towards the midline and separated by nasals only). The analysis was run in MrBayes 3.2.6, on the Cyberinfrastructure for Phylogenetic Research (CIPRES) Science Gateway (Miller, Pfeiffer & Schwartz, 2010). Our new morphological codings and the full matrix are available from MorphoBank, project 2225 (full matrix stored in the “Documents” section) and as part of Supplemental Information 1.

Age determination

To determine the age of the new specimen, we searched a sample of in situ sediment recovered from the cranium for biostratigraphically informative palynomorphs. The extraction procedure followed the standard protocol of Louwye et al. (2007), and involved successive treatments with HCl and HF to remove carbonates and silicates, respectively. No oxidation or ultrasonic treatment was applied to avoid damage and selective loss of species. The organic residue was mounted with glycerine jelly on two microscope slides, which were then systematically scanned for palynomorphs. Nomenclature of the dinoflagellate cysts follows Fensome, MacRae & Williams (2008).

Nomenclatural acts

The electronic version of this article in Portable Document Format (PDF) will represent a published work according to the International Commission on Zoological Nomenclature (ICZN), and hence the new names contained in the electronic version are effectively published under that Code from the electronic edition alone. This published work and the nomenclatural acts it contains have been registered in ZooBank, the online registration system for the ICZN. The ZooBank LSIDs (Life Science Identifiers) can be resolved and the associated information viewed through any standard web browser by appending the LSID to the prefix http://zoobank.org/. The LSID for this publication is: urn:lsid:zoobank.org:pub:E728C3DD-EB85-482F-ACE6-6558E3ED5441. The online version of this work is archived and available from the following digital repositories: PeerJ, PubMed Central and CLOCKSS.

Results

Systematic palaeontology

Cetacea Brisson, 1762 sensu Geisler et al. (2011)	
Neoceti Fordyce and Muizon, 2001 sensu Geisler et al. (2011)	
Mysticeti Gray, 1864 sensu Geisler et al. (2011)	
Chaeomysticeti Mitchell, 1989 sensu Geisler et al. (2011)	
Cetotheriidae Brandt, 1872; sensu Fordyce & Marx (2013)	
Metopocetus Cope, 1896	

Type species. Metopocetus durinasus Cope, 1896

Emended diagnosis. Small to medium-sized cetotheriid differing from all other chaeomysticetes except cetotheriids in having a distally expanded compound posterior process of the tympanoperiotic bearing a floored facial sulcus, as well as medially convergent ascending processes of the maxillae bearing an enlarged, primary dorsal infraorbital foramen (new term); further differs from all other chaeomysticetes except cetotheriids and balaenopterids in having the ascending process of the maxilla and the parietal overlap anteroposteriorly; and from balaenopterids in having the apex of the supraoccipital shield located posterior to the supraorbital process of the frontal. Differs from other cetotheriids, including neobalaenines, in lacking a well-developed lateral tuberosity of the periotic, and in having a better-defined mallear fossa and a well-developed paroccipital concavity and tympanohyal; from all other cetotheriids, except possibly Joumocetus Kimura & Hasegawa, 2010, in having a distinctly triangular ascending process of the maxilla; from Herpetocetus, Nannocetus, Cephalotropis Cope, 1896 and neobalaenines in having the posterior portion of the zygomatic process of the squamosal offset from the lateral border of the exoccipital by a distinct angle; from Herpetocetus, Nannocetus and Piscobalaena Pilleri and Siber, 1989 in the presence of a squamosal cleft; from Herpetocetus and Nannocetus in having a smaller temporal exposure of the alisphenoid and in having a transversely oriented postglenoid process; from Brandtocetus Gol’din & Startsev, 2014, Cetotherium, Joumocetus, Kurdalagonus Tarasenko & Lopatin, 2012, “Aulocetus” latus, “Cetotherium” megalophysum, “Metopocetus” vandelli and likely also Herentalia Bisconti, 2014 in having a (slightly) more plug-like compound posterior process of the tympanoperiotic; from Brandtocetus, Cephalotropis, Cetotherium, Joumocetus, Kurdalagonus, Vampalus Tarasenko & Lopatin, 2012, Zygiocetus Tarasenko, 2014, “Aulocetus” latus, “Cetotherium” megalophysum and “Metopocetus” vandelli in having a more rounded apex of the supraoccipital shield; from Brandtocetus, Cetotherium and Zygiocetus in having a tympanic bulla that is not transversely wider anteriorly than it is posteriorly; and from Joumocetus and Cephalotropis in having the parietal almost excluded from the intertemporal region.

Metopocetus hunteri, sp. nov.

Figs. 2–9

LSID. urn:lsid:zoobank.org:act:391CF6D9-138C-4F88-AC4B-9903DA433FDA.

Holotype. NMR 9991-07729, a partial cranium preserving the vertex, palatines, the right half of the braincase and basicranium, and the right periotic and tympanic bulla.

Locality and horizon. Sand pit at Liessel, Deurne, North Brabant, the Netherlands (Fig. 1). The coordinates of the type locality are N51°25′44″E5°49′47″. The specimen was retrieved from deposits assigned to the Breda Formation, a shallow marine unit consisting of glauconiferous sands, sandy clays and clays. The Breda Formation is widespread throughout the Netherlands and comprises the greater part of the Dutch Miocene succession (Burdigalian–Tortonian), reaching as much as 700 m in thickness in some locations (Munsterman & Brinkhuis, 2004).

Figure 2 Cranium in dorsal and posterolateral view.

Cranium of Metopocetus hunteri in (A) dorsal and (B) posterolateral view.

The preservation of the dinoflagellate cyst assemblage recovered from the matrix associated with the specimen is moderate to good. In total, we recorded 28 dinoflagellate cyst species and three acritarchs (Table S1), the most important of which include Barssidinium taxandrianum Louwye, 1999, Gramocysta verricula (Piasecki, 1980), Habibacysta tectata Head et al., 1989, Hystrichosphaeropsis obscura Habib, 1972 and Labyrinthodinium truncatum Piasecki, 1980. H. tectata first occurs in the North Atlantic realm (Porcupine Basin, off southwest Ireland) during the Langhian, around 14.2 Ma (Hilgen, Lourens & Van Dam, 2012; Louwye et al., 2008; Quaijtaal et al., 2014), thus setting a maximum age for the sample. Conversely, the minimum age is determined by the highest occurrences of Hy. obscura and L. truncatum at approximately 7.6 Ma (De Verteuil & Norris, 1996; Dybkjær & Piasecki, 2010; Köthe, 2012; Louwye & De Schepper, 2010; Munsterman & Brinkhuis, 2004).

The sample belongs to the late Tortonian (Late Miocene) SNSM14 Zone defined in the Netherlands (Munsterman & Brinkhuis, 2004), which is equivalent to the Hystrichosphaeropsis obscura biozone of Denmark (Dybkjær & Piasecki, 2010), and the DN9 Zone of the eastern USA and Germany (De Verteuil & Norris, 1996; Köthe, 2012), dated to ca 8.8–7.6 Ma (Dybkjær & Piasecki, 2010). The upper boundary of the SNSM14 Zone is defined by the highest occurrence of L. truncatum, while the lower boundary is defined by highest occurrence of Cleistosphaeridium placacanthum Deflandre and Cookson, 1955, a distinctive dinoflagellate cyst species not recorded in our sample. Diagnostic species present in this zone are G. verricula and Hy. obscura (Munsterman & Brinkhuis, 2004). Further evidence for this age assessment comes from the occurrence of B. taxandrianum, which is a rare species with a restricted occurrence in the Late Miocene of the southern North Sea Basin, including the Tortonian Diest and the latest Tortonian–Messinian Kasterlee Formations (Louwye, 1999; Louwye & de Schepper, 2010; Louwye et al., 2007; Louwye & Laga, 2008). This species has never been recorded from Pliocene deposits.

Besides age determination, the recovered dinoflagellates also provide some insights into the depositional environment. In this context, the presence of Gramocysta verricula is particularly notable. This species was first recorded from the Late Miocene Gram Formation of Denmark, where it dominates the eponymous biozone (Piasecki, 1980). The latter is furthermore characterised by the disappearance of neritic genera, such as Achomosphaera Evitt, 1963 and Tectatodinium Wall, 1967, and an overall reduction in the abundance of other dinocyst species. Together, these events likely reflect a marine regression, accompanied by high sedimentation rates and an enhanced influx of freshwater (Piasecki, 1980). The preference of G. verricula for marginal marine environments is further corroborated by its occurrence in the shallow marine Kasterlee Formation and other deposits recording marked drops in sea level (Louwye et al., 2007).

Etymology. Named after the famous Scottish surgeon and anatomist John Hunter, who was maybe the first person to recognise and write about the similarity of whales and artiodactyls (Hunter, 1787).

Diagnosis. Differs from Metopocetus durinasus in having a somewhat narrower, less distally exposed compound posterior process of the tympanoperiotic, a less anteriorly bulging temporal wall of the squamosal and a more proximally located primary dorsal infraorbital foramen on the ascending process of the maxilla (located either more distally or absent in M. durinasus), as well as in lacking ankylosed nasals.

Description

Overview. The preserved, mostly right portion of the cranium lacks both the rostrum and the supraorbital process of the frontal (Fig. 2). The apex of the zygomatic process, the central portion of the nuchal crest, the tip of the postglenoid process and much of the right pterygoid are broken. The state of preservation of the bones that remain is relatively good, but a certain degree of surface damage and small pockets of remaining matrix (e.g., on the dorsal surface of the periotic) sometimes make it difficult to discern details. Measurements of the cranium are shown in Table 1.

Table 1 Measurements of Metopocetus hunteri (in mm).

Cranium excluding ear bones	
Maximum length of right nasal, as preserved	137.0+	
Maximum length of left nasal, as preserved	155.0	
Anteroposterior diameter of primary dorsal infraorbital foramen	27.0	
Transverse diameter of primary dorsal infraorbital foramen	14.5	
Length of sulcus continuing posteriorly from primary dorsal infraorbital foramen	13.0	
Length of slit-like sulcus on ascending process of maxilla, anteromedial to primary dorsal infraorbital foramen	25.0	
Minimum transverse width across parietals on vertex	30.5	
Maximum distance between sagittal plane and outer surface of the zygomatic process, as preserved	285.0	
Maximum distance between sagittal plane and lateral border of exoccipital	190.0	
Anteroposterior length of pterygoid sinus fossa	64.0	
Transverse width of pterygoid sinus fossa	56.0	
Transverse width of postglenoid process at base	124	
Maximum diameter of foramen pseudovale	20.0	
Distance from posteromedial corner of falciform process of squamosal to innermost portion of internal acoustic meatus	32.5	
Anteroposterior diameter of external acoustic meatus	28.0	
Transverse width of basioccipital crest	47.0	
Transverse width of jugular notch	10.7	
Maximum anteroposterior diameter of paroccipital concavity	60.0	
Maximum transverse diameter of paroccipital concavity	56.0	
Maximum height of foramen magnum	51.0a	
Maximum height of right occipital condyle	87.0	
Maximum width of right occipital condyle	47.2	
Bicondylar widtha	150.0	
Periotic and tympanohyal	
Anteroposterior length of anterior pedicle	9.0	
Maximum anteroposterior width of pars cochlearis, measured up to the medial border of the fenestra rotunda	18.6	
Maximum diameter of fenestra rotunda	5.8	
Maximum diameter of proximal opening of facial canal	7.0	
Maximum diameter of dorsal vestibular area	7.0	
Maximum diameter of aperture for cochlear aqueduct	4.0	
Maximum anteroposterior diameter of facial sulcus	9.7	
Maximum dorsoventral diameter of facial sulcus	11.5	
Anteroposterior length of lateral exposure of compound posterior process	33.0	
Maximum proximodistal length of tympanohyal	22.3	
Maximum diameter of distal surface of tympanohyal	7.3	
Tympanic bulla and malleus	
Maximum anteroposterior length of tympanic bulla	77.1	
Anteroposterior length of dorsal aperture of tympanic cavity	56.0	
Width of bulla just anterior to the sigmoid process	47.3	
Transverse width of sigmoid process	17.3	
Transverse width of conical process	8.1	
Maximum length of posterior pedicle	16.7	
Maximum diameter of malleus, from the head to the tip of the tubercule	11.7	
Maximum dorsoventral height of head of malleus	7.6	
Notes.

a Estimated

Maxilla, premaxilla and nasal. Of the maxilla, only the triangular ascending process is preserved, which extends posteriorly beyond the base of the supraorbital process of the frontal and overlaps with the parietal (Figs. 2 and 3). In cross section, the ascending process is markedly concave, with its medial border rising towards the nasal. Medially, the apices of the ascending processes are clearly convergent, but remain separated from each other by the well-developed nasals. Near the base of the ascending process, there is a large primary dorsal infraorbital foramen (new term), which is also found in other cetotheriids and exits into a short, dorsomedially oriented sulcus (Fig. 3B). Anteromedial to this foramen, there are two elongate sulci without obvious foramina running parallel to the medial margin of the maxilla. Inside the narial fossa, the maxilla gives rise to a narrow shelf supporting the anterolateral corner of the nasal.

Figure 3 Temporal fossa and vertex.

Detail of the cranium of Metopocetus hunteri: (A) posteromedial wall of temporal fossa in anterolateral view; (B) vertex in anterodorsal view.

Nothing remains of the premaxilla, but the close juxtaposition of the posterior portions of the nasals and maxillae suggests that it did not extend as far posteriorly as the other rostral bones; instead, it likely terminated somewhere along the anterior half of the nasal, as in Herpetocetus and, presumably, Piscobalaena, “Cetotherium” megalophysum and “Metopocetus” vandelli (El Adli, Deméré & Boessenecker, 2014). In dorsal view, the nasal is anteroposteriorly elongate and somewhat triangular, with its lateral and medial borders converging posteriorly (Fig. 3B). Although transversely narrow posteriorly, it is exposed on the cranial vertex along its entire length—unlike in Herpetocetus and Piscobalaena, in which the posterior portion of the nasal is nearly invisible. The anterior portions of both nasals are eroded, but seem to have formed a straight or slightly convex anterior border, without any obvious sagittal crest or anterior projection as in Herpetocetus, Piscobalaena and neobalaenines.

Frontal. Only the portion of the frontal supporting the ascending process of the maxilla is preserved (Fig. 2). In dorsal view, the frontal is almost entirely excluded from the cranial vertex by the maxilla, but still overrides much of the anterior portion of the parietal. Laterally, the posterior margin of the frontal gradually descends anteroventrally towards the base of the supraorbital process of the frontal. In lateral view, the dorsal portion of the fronto-parietal suture is elevated into a ridge slightly overhanging the anteriormost portion of the parietal (Fig. 3A), as also seen in Herentalia and Piscobalaena.

Parietal. In dorsal view, the parietal is exposed as a thin band on the vertex, anterior to the apex of the supraoccipital shield (Fig. 3B). Anteroventral to the vertex, the parietal becomes markedly concave as it descends towards the base of the supraorbital process of the frontal. In lateral view, the parietal is slightly longer anteroposteriorly than high dorsoventrally (Fig. 4A). The parieto-squamosal suture is smooth, with no obvious hint of a ridge-like eminence or a tubercle at the point where the suture meets the nuchal crest. Unlike in Herpetocetus, there is no postparietal foramen (Fig. 3A).

Figure 4 Cranium in lateral and posterior view.

Cranium of Metopocetus hunteri in (A) lateral and (B) posterior view.

Alisphenoid. The alisphenoid is exposed in the temporal fossa and contacts the parietal, the squamosal and the pterygoid. In lateral view, the preserved portion of the alisphenoid is nearly circular in outline and relatively large (Fig. 3A)—larger than in Cetotherium riabinini and comparable to that of “Cetotherium” megalophysum, but still much smaller than in Herpetocetus (El Adli, Deméré & Boessenecker, 2014; Gol’din, Startsev & Krakhmalnaya, 2014). Anteroventrally, the alisphenoid likely contributed to the rim of the orbital fissure. In ventral view, the alisphenoid is covered by the dorsal lamina of the pterygoid.

Squamosal. In dorsal view, the temporal surface of the squamosal is relatively even and does not markedly bulge into the temporal fossa. The posterior border of the temporal fossa is smooth with no squamosal crease (Fig. 2A). There is a well-developed squamosal cleft that originates at the parieto-squamosal suture and runs towards the base of the zygomatic process (Fig. 3A); a similar cleft occurs in Cephalotropis and “Cetotherium” megalophysum. The squamosal fossa is anteroposteriorly elongate, with its floor being convex anteriorly, but concave posteriorly as it approaches the posterior apex of the nuchal crest. The zygomatic process is broken, but has a robust base bearing a distinct supramastoid crest and, unlike Piscobalaena, “Cetotherium” megalophysum and herpetocetines, a small squamosal prominence (Fig. 2B). Judging from what remains, the zygomatic process seems to have been oriented anteriorly. Posteriorly, the zygomatic process is laterally offset from the rest of the cranium (unlike in herpetocetines and Caperea), with its posterior border forming a 90° angle with the lateral margin of the exoccipital and the portion of the squamosal surrounding the periotic (Fig. 2A).

In lateral view, there is a well-defined sternomastoid fossa (sensuBouetel & De Muizon, 2006) located just ventral to the supramastoid crest (Figs. 2B and 4A). The preserved portion of the postglenoid process is triangular in outline and points slightly posteroventrally. The base of the zygomatic process is robust. In posterior view, the postglenoid is parabolic in outline and seems to point directly ventrally, rather than medially as in herpetocetines, although its exact shape it lost owing to breakage (Fig. 4B). The posterior meatal crest extends from the external acoustic meatus on to the posterior face of the postglenoid process, where it forms a well-developed horizontal shelf. In doing so, it defines a deep sulcus running parallel to the meatus, immediately below the sternomastoid fossa (Figs. 2B and 4B).

In ventral view, the falciform process of the squamosal is robust, distinctly squared and, along with adjacent portions of the squamosal, forms virtually the entire rim of the foramen pseudovale (Fig. 5). The external acoustic meatus is relatively broad, with its roof—the posterior meatal crest—extending on to the anterior face of the posterior process of the periotic. Together with the falciform process, the innermost portion of the internal acoustic meatus defines a strikingly rectangular window exposing the lateral surface of the anterior process of the periotic (Fig. 6A). Anterior to the meatus, the postglenoid process of the squamosal is thin anteroposteriorly, oriented transversely and medially confluent with the anterior meatal crest.

Figure 5 Cranium in ventral view.

Cranium of Metopocetus hunteri in ventral view.

Figure 6 Basicranium and periotic.

Basicranium and periotic of Metopocetus hunteri: (A) right portion of basicranium in ventral view; (B) central portion of periotic in ventromedial view; (C) compound posterior process of tympanoperiotic in external view; (D) central portion of periotic in dorsal view. Abbreviations: am, anteromedial; ant, anterior; fac., facial sulcus; lat, lateral; parocc. conc., paroccipital concavity; pm, posteromedial; pos, posterior; post. process, compound posterior process; ven, ventral.

Supraoccipital. In dorsal view, the supraoccipital shield is broadly triangular, with a straight to slightly convex lateral border (=nuchal crest) and a rounded apex (Fig. 2A). As in all other cetotheriids except neobalaenines, the nuchal crest is oriented mostly dorsally and does not overhang the temporal fossa. Just posterior to the apex of the supraoccipital shield, there is a relatively broad, tabular area that posteriorly gives rise to an external occipital crest. The latter is well-developed and extends along at least one third of the dorsal surface of the supraoccipital; further posteriorly, the central portion of the bone is missing (Fig. 2). In posterior view, the supraoccipital is markedly concave transversely, without any obvious tubercles on either side of the external occipital crest (Fig. 4B).

Exoccipital and basioccipital. In dorsal view, the exoccipital is well developed and extends posteriorly beyond both the level of the occipital condyle and the posterior apex of the nuchal crest (Fig. 2). The occipital condyle is large and situated on a distinct neck. In posterior view, the paroccipital process is squared in outline and extends ventrally to roughly the same level as the basioccipital crest (Fig. 4B). Medial to the paroccipital process, the jugular notch is narrow transversely and elongate dorsoventrally. The foramen magnum is framed by the dorsal portion of the occipital condyle.

In ventral view, the entire ventral surface of the exoccipital is excavated by the paroccipital concavity (Fig. 6A). Medially, this fossa invades, and is thus partially floored by, the ventromedial corner of the paroccipital process, which also separates it from the jugular notch. Laterally, the paroccipital concavity is relatively open. Anteriorly, the floor of the paroccipital concavity forms a shelf that partially floors the facial sulcus, and is in turn underlapped by a posteroventral flange (new term) arising from the compound posterior process of the tympanoperiotic (Figs. 6A and 6C). This contact between the exoccipital and the posteroventral flange of the tympanoperiotic—which, to our knowledge, is unique among mysticetes—creates a continuous bony surface that allows the paroccipital concavity to extend far on to the tympanoperiotic itself (Figs. 5, 6A and 6C). Medial to the well-marked jugular notch, the basioccipital crest is transversely broad, triangular and oriented anteroposteriorly (Fig. 5). As far as can be told, the suture between the basioccipital and the basisphenoid is ventrally covered by the posteriormost portion of the vomer.

Vomer. Only the posterior portion of the vomer is preserved. In the basicranium, the vomer is broadly exposed posterior to what remains of the choanae and overrides much of the medial lamina of the pterygoid. Further anteriorly, the vomer is exposed between the anterior portions of the palatines, as in all other cetotheriids for which the condition of this part of the vomer is known (Fig. 5).

Palatine. Both palatines are preserved, but have lost nearly all of their outer margins; they are markedly concave transversely, as if pinched, thus forming a distinct ventral keel. A similar condition occurs in Cephalotropis, Caperea and, to some degree, Herpetocetus. By contrast, the palatines are only slightly concave in Piscobalaena, “Cetotherium” megalophysum and “Metopocetus” vandelli, and seemingly flattened or even slightly convex in Cetotherium (Gol’din, Startsev & Krakhmalnaya, 2014).

Pterygoid. The ventral portion of the pterygoid is mostly missing, except for a small portion contributing to the rim of the foramen pseudovale. Dorsally, the pterygoid roofs almost the entire pterygoid sinus fossa, which extends anteriorly approximately to the level of the foramen pseudovale. Posteriorly, the dorsal or lateral lamina of the pterygoid overrides the anteriormost portion of the anterior process of the periotic (Figs. 5 and 6A). Medially, the pterygoid is continuous with the basioccipital crest.

Periotic, stapes and tympanohyal. In ventral view, the anterior process of the periotic appears to be transversely thickened, but not hypertrophied (Fig. 6A). The lateral tuberosity is indistinct, in stark contrast to herpetocetines and, to a lesser degree, Brandtocetus and Kurdalagonus. The anterior pedicle is relatively small and located just anterior to the broad and comparatively well-defined mallear fossa. There is no anterior bullar facet, and seemingly no distinct ridge for the attachment of the tensor tympani muscle, unlike in herpetocetines and Piscobalaena. The pars cochlearis is rounded and posteriorly terminates in an elongate caudal tympanic process, which approaches, but does not contact, the crista parotica (Fig. 6B). The presence or absence of the promontorial groove is unclear. Sediment obscures both the distal opening of the facial canal and the fenestra ovalis, but the ventral portion of the right stapes can be seen to protrude from the latter.

The compound posterior process of the tympanoperiotic (hereafter shortened to posterior process) is oriented posterolaterally relative to the anteroposterior axis of the pars cochlearis. At its base, it carries the posterior pedicle of the tympanic bulla, which appears curved as a result of internal excavation by the tympanic cavity (Figs. 6A and 6B). Next to the posterior pedicle, there is a large, trumpet-shaped tympanohyal fused to the crista parotica (Fig. 6B). The presence of such a well-developed tympanohyal is rare among mysticetes, and among cetotheriids only occurs in Metopocetus. Along its anterior margin, the posterior process gives rise to a posteriorly excavated anteroventral flange (new term), which anteriorly delimits the expanded paroccipital concavity (Figs. 6A and 6C). The floor of the paroccipital concavity is formed by a horizontal posteroventral flange (new term) that underlaps both the facial canal and the anterior rim of the ventral surface of the exoccipital (Figs. 6A and 6C).

In medial view, the anterior process appears two-bladed, but its actual shape is difficult to discern because it is partially covered by the dorsal/lateral lamina of the pterygoid. The fenestra rotunda is large and offset from the posterior border of the pars cochlearis by a broad shelf (Fig. 6B). Ventrally, this shelf merges with the elongate, posteriorly oriented caudal tympanic process. In dorsal view, the dorsal vestibular area and the proximal opening of the facial canal are comparable in size and separated by a well-developed transverse septum (Fig. 6D). Together, they are nearly, albeit not perfectly, in line with the circular aperture for the cochlear aqueduct. The aperture for the vestibular aqueduct is obscured by matrix, but does not seem to overlap anteroposteriorly with the aperture for the cochlear aqueduct. The suprameatal fossa is shallow with a rounded lateral border; there is no distinct superior process. In lateral view, the posterior process is broadly exposed on the lateral skull wall, but anteroposteriorly narrower than in Metopocetus durinasus and herpetocetines (Fig. 6C) (Whitmore & Barnes, 2008). The facial sulcus runs along the posterior border of the posterior process. Just anterior to the distal opening of the facial sulcus, the outermost portion of the anteroventral flange is expanded and ventrally delimits a deep fossa of unknown function and homology (Fig. 6C).

Tympanic bulla. In dorsal view, the involucrum is relatively narrow in the area of the anteroposteriorly broad Eustachian outlet, but then rapidly widens as it approaches the posterior pedicle (Figs. 7A and 8A). There are no obvious transverse sulci on its dorsal surface, except for some rims in the vicinity of the posterior pedicle. Transverse sulci are common in mysticetes and marked in adult specimens of at least some cetotheriids (e.g., Brandtocetus chongulek and Herpetocetus transatlanticus). It is possible that their absence in NMR 9991-07729 is a result of surface damage, although it seems likely that even in a perfectly preserved bulla they would have been at best faintly developed. A smooth involucrum is typical of juvenile individuals, and may hence indicate that NMR9991-07729 is more likely to be an old juvenile than an adult.

Figure 7 Tympanic bulla—photographs.

See Fig. 8 for explanatory line drawings. Abbreviations: ant, anterior; dl, dorsolateral; dor, dorsal; lat, lateral; med, medial; pm, posteromedial; pos, posterior.

Figure 8 Tympanic bulla—explanatory line drawings.

The involucral ridge (sensu Oishi & Hasegawa, 1995) extends all the way to the medial margin of the bulla, largely as a result of the robustness of the inner posterior prominence (=medial lobe of the tympanic bulla). The sigmoid process is oriented transversely and situated roughly halfway along the anteroposterior length of the bulla; its dorsomedial corner is distinct from the anterior process of the malleus and twisted slightly posteriorly. The conical process is transversely thickened and located entirely posterior to the sigmoid process. Opposite the conical process, the posterior pedicle is located relatively close to the posterior border of the bulla and internally excavated by a branch of the tympanic cavity. In medial view, the bulla is somewhat pear-shaped in outline, with the dorsal surface of the involucrum being distinctly concave (Fig. 7B and Fig. 8B). In the region of the Eustachian outlet, the dorsal surface of the involucrum is depressed into a broad, smooth fossa. The main and involucral ridges converge anteriorly, while being more clearly separated posteriorly by a relatively shallow median furrow and interprominential notch. On the medial face of the conical process, the tympanic sulcus follows a broad, horizontal ridge somewhat similar to that in Piscobalaena, before suddenly turning 90° to run dorsally on to the posterior surface of the sigmoid process (Figs. 7G and Fig. 8G).

In ventral view, the anterior portion of the bulla appears to be more rounded than in most other cetotheriids, although the anterior border is still somewhat flattened (Figs. 7C and 8C). There is no anterolateral shelf. The anterolateral corner of the bulla is inflated and forms a distinct lobe anterior to the lateral furrow. The outline of the main ridge (sensu Oishi & Hasegawa, 1995) is convex. In lateral view, the lateral furrow is distinct and oriented vertically (Figs. 7D and 8D). The sigmoid cleft ventrally merges into the outer surface of the bulla, so that there is no discernable ventral border of the sigmoid process. Consequently, the latter does not overlap the anterior portion of the conical process, although the two processes are still connected by a well-developed horizontal rim. The conical process itself is dorsally rounded, not flattened as in Herpetocetus and Caperea.

In anterior view, the ventral surface of the bulla is transversely convex, except for a small concave portion immediately medial to the main ridge (Figs. 7E and 8E). The rim of the Eustachian outlet is oriented horizontally and continuous with the dorsal surface of the involucrum. The lateral margin of the sigmoid process is oriented slightly dorsolaterally, but the process as a whole is not laterally deflected. In posterior view, the main ridge of the bulla is oriented medially, so that the inner posterior prominence faces dorsally, and the outer posterior prominence ventrally (Figs. 7F and 8F). Like most other chaeomysticetes, the bulla thus shows a marked degree of medial rotation relative to the condition in archaic toothed mysticetes and eomysticetids. The involucral ridge is well developed and terminates ventral to the base of the posterior pedicle. There is neither a transverse crest connecting the main and involucral ridges, nor an elliptical foramen. The lateral margin of the conical process is straight.

Malleus. In posterodorsal view, the articular facets for the incus are oriented at right angles to each other, with the vertical facet being slightly larger (Fig. 9). The head of the malleus is broadly rounded and separated from the tubercule by a distinct groove. In anterior view, the bottom of the head and the anterior process are excavated by the sulcus for the chorda tympani. Adjacent to the internal margin of the head, the muscular process bears a well-defined, circular pit for the insertion of the tendon of the tensor tympani muscle (Fig. 9).

Figure 9 Malleus.

Malleus of Metopocetus hunteri in (A) posterior and (B) anterior view. Abbreviations: dor, dorsal; med, medial; ven, ventral.

Discussion and Conclusions

Age of Metopocetus

The age of the hitherto only member of Metopocetus, M. durinasus, has been a matter of some debate. In his original description of the holotype and only specimen of M. durinasus, USNM 8518, Cope (1896) provided little detail as to the provenance of the material, stating only that it had been collected from “a Miocene marl from near the mouth of the Potomac river” (p. 143). Subsequent authors interpreted this description of the type locality to refer to either the Calvert Formation (Kellogg, 1931; Kellogg, 1968) or the St. Mary’s Formation (Case, 1904), implying either a Langhian or a Tortonian age, respectively (Marx & Fordyce, 2015). Determining which of these possibilities is correct is crucial, given that a Langhian age would make M. durinasus the oldest reported cetotheriid. The occurrence of M. hunteri in Tortonian strata of Europe suggests that M. durinasus may also date from this stage, especially given the relatively close morphological resemblance of the two species. This idea is furthermore consistent with the occurrence of at least two other cetotheriids (Cephalotropis coronatus and “Cetotherium” megalophysum) in the St Mary’s Formation, whereas the family is conspicuously absent from the Calvert Formation. Pending the discovery of additional specimens and/or direct dating evidence, we thus suggest that M. durinasus should likely be regarded as Tortonian.

Ontogenetic age

Except for those of the maxillae and nasals, all of the cranial sutures are closed, which suggests that this individual is at or near its adult size. Support for this estimate comes from the presence of several well-developed bony crests, such as the external sagittal crest on the supraoccipital, the supramastoid crest on the squamosal and a reasonably distinct main ridge on the tympanic bulla. The anteroventral displacement of the maxillae and nasals does not necessarily contradict this assessment, as in modern mysticetes the sutures connecting these bones tend to be relatively loose even in adults to facilitate rostral kinesis (e.g., Deméré & Berta, 2008). Potentially more problematic is the rather smooth texture of the dorsal surface of the involucrum, which is typical of juveniles. Some of this smoothness may be due to superficial damage, but there is no evidence that the original texture of the involucrum markedly differed from what is preserved. In the absence of more definitive markers of development, such as vertebral or long bone epiphyses, it thus seems most consistent to interpret the present material as a relatively old juvenile.

Phylogeny

Our phylogenetic analysis (average deviation of split frequencies 0.016 after 50 million generations) clearly places Metopocetus hunteri inside both Cetotheriidae and as sister to M. durinasus (Fig. 10). “Metopocetus” vandelli is not closely related to either M. durinasus or M. hunteri, and instead clusters with “Aulocetus” latus and “Cetotherium” megalophysum. Beyond this, our results largely correspond to those of Marx & Fordyce (2015), but differ in two important aspects: (1) Metopocetus is no longer grouped with Piscobalaena and “C.” megalophysum, and instead now forms part of a basal lineage along with Cephalotropis; (2) Piscobalaena and “C.” megalophysum no longer cluster with Cetotherium and instead now form a clade with Herpetocetinae + Neobalaeninae.

Figure 10 Phylogenetic relationships of Metopocetus hunteri, based on a dated total evidence analysis.

All data except the codings for M. hunteri, “M.” vandelli and “Aulocetus” latus are from Marx & Fordyce (2015: Fig. 2). Drawings of cetaceans by Carl Buell. Abbreviations: Pli., Pliocene; Pls., Pleistocene.

Cephalotropis has previously been found to occupy a basal position within Cetotheriidae (El Adli, Deméré & Boessenecker, 2014; Gol’din & Steeman, 2015), which is at least partially reflected by our results. Nevertheless, the grouping of Metopocetus and Cephalotropis is novel and somewhat surprising, given their superficially rather different morphologies. This discrepancy is reflected in the low posterior probability (<50%) of the node that unites them, as well as the considerable length of the branch leading to Cephalotropis. The clade is supported by the presence of a well-developed median keel on the palatines (char. 22), but it is worthwhile noting that a similar morphology also occurs in neobalaenines and, up to a point, Herpetocetus.

The move of Piscobalaena closer to herpetocetines is less controversial than the grouping of Cephalotropis and Metopocetus, and brings our findings into line with those of several earlier studies (Bisconti, 2015; El Adli, Deméré & Boessenecker, 2014; Gol’din & Startsev, 2014; Gol’din & Steeman, 2015). Nevertheless, the branch uniting Piscobalaena + “Cetotherium” megalophysum with herpetocetines + neobalaenines has a low posterior probability, even though it is supported by 3 synapomorphies: an orbitotemporal crest running close to the posterior border of the supraorbital process (char. 80); absence of the squamosal prominence (char. 106); and presence of a sulcus marking the attachment of the mylohyoid muscle on the inside of the mandible (char. 238). Considerably better supported relationships, though not of immediate interest to this study, include the clade comprising the Cetotherium-like taxa (Brandtocetus, Cetotherium and Kurdalagonus) from the Eastern Paratethys, neobalaenines, herpetocetines, and the branch uniting the latter two (Fig. 10).

The relatively basal position of Metopocetus is inconsistent with it showing a morphology truly intermediate between that of herpetocetines and other cetotheriids (Whitmore & Barnes, 2008). It furthermore implies that the pronounced widening of the distal portion of the compound posterior process—a hallmark of cetotheres—may have occurred more than once. The posterior process of all cetotheriids is large relative to that of most other mysticetes, but there are clear differences in scale: its distal end is most expanded in herpetocetines, neobalaenines, Cephalotropis, M. durinasus and Piscobalaena; somewhat less so in Brandtocetus, Cetotherium, Kurdalagonus, M. hunteri and Zygiocetus; and even less so in “Aulocetus” latus, “C.” megalophysum and “M.” vandelli.

“Cetotherium” megalophysum and “Metopocetus” vandelli were included in Herpetocetinae as sister to Nannocetus by El Adli, Deméré & Boessenecker (2014), whereas “C.” megalophysum fell out as sister to Piscobalaena in the present analysis. Both topologies require that the distal widening of the posterior process either occurred in parallel in several lineages, or else was later reduced in certain species. The topology of Gol’din & Steeman (2015) partially circumvents this problem by excluding “C.” megalophysum and “M.” vandelli from Cetotheriidae altogether, but even in this case widening of the posterior process would have occurred at least twice: once in the lineage leading to Cephalotropis and neobalaenines, and once within their Cetotheriidae proper. There is, of course, a distinct possibility that this patchy character distribution is simply the result of errors in the cladistic hypotheses. Nevertheless, given the wide range of morphologies and generally mosaic distribution of characters within Cetotheriidae, we suggest that the presence of an expanded posterior process may reflect a shared evolutionary trend within the family, rather than a definitive uniting character. A better understanding of the history of this unique feature will likely depend on getting to grips with its function first.

In addition to that of M. hunteri, the position of “Metopocetus” vandelli is of particular interest to the present study, as it is the only other species ever referred to Metopocetus (Kellogg, 1941). Recent analyses have cast considerable doubt on this assignment, and variously grouped “M. ” vandelli with “C.” megalophysum, a clade comprising Piscobalaena, Metopocetus and herpetocetines, or even an entirely different family, Tranatocetidae, thought to be related to balaenopterids and eschrichtiids (El Adli, Deméré & Boessenecker, 2014; Gol’din & Startsev, 2014; Gol’din & Steeman, 2015). Even at a relatively cursory glance, “M.” vandelli clearly differs from both M. durinasus and M. hunteri in a range of features, including (1) a more elongate, finger-like ascending process of the maxilla; (2) a more pointed, dorsally flattened supraoccipital shield lacking a well-developed external occipital crest; (3) the apparent absence of a squamosal cleft (not completely clear owing to incomplete preparation of the type specimen); (4) comparatively flat palatines not forming a medial ridge; (5) a markedly less expanded distal portion of the compound posterior process (to be confirmed by further preparation); and (6) a more gracile exoccipital (Fig. 11). Taken together, these differences speak against any particularly close affinity of “M.” vandelli with Metopocetus and thus support its removal from this genus, as advocated by several other recent studies (El Adli, Deméré & Boessenecker, 2014; Gol’din & Startsev, 2014; Gol’din & Steeman, 2015; Whitmore & Barnes, 2008).

Our analysis agrees with previous studies in grouping “A.” latus and “M.” vandelli into a clade with “C.” megalophysum (El Adli, Deméré & Boessenecker, 2014; Gol’din & Steeman, 2015). Support for this branch is reasonable at 89%, although we currently only recognise a single synapomorphy: the posterior projection of the occipital condyles beyond the level of the exoccipitals (char. 139). A more detailed examination of this proposed relationship is beyond the scope of this study, and furthermore currently hampered by the incomplete preparation and lack of description of the available material. Nevertheless, in light of the consistency with which these taxa have been grouped together in recent analyses, we tentatively suggest that all of them may not only be closely related, but possibly even congeneric or conspecific. Further data, especially on the morphology of the ear bones, will provide the means to test this idea.

Paroccipital concavity

Metopocetus stands out for having an unusually enlarged paroccipital concavity extending across both the exoccipital and the compound posterior process of the tympanoperiotic (Fig. 6A). A fossa excavating the anteroventral surface of the paroccipital process occurs in a variety of cetaceans, including archaeocetes, mysticetes and odontocetes (e.g., Deméré & Berta, 2008; Fraser & Purves, 1960; Martínez Cáceres & De Muizon, 2011). Among mysticetes, the paroccipital concavity tends to be best developed in archaic forms and least in the extant taxa (e.g., Deméré & Berta, 2008; El Adli, Deméré & Boessenecker, 2014). Nevertheless, its size and shape is variable, and the concavity remains well-developed in at least one living species, the grey whale, Eschrichtius robustus (Lilljeborg, 1861) (Fig. 12). In terms of its function, the paroccipital concavity is generally interpreted as the bony correlate of the posterior sinus and/or the site of the ligamentous attachment of the stylohyal to the basicranium (Beauregard, 1894; Boessenecker & Fordyce, 2015; Bouetel & De Muizon, 2006; Deméré & Berta, 2008; El Adli, Deméré & Boessenecker, 2014; Fraser & Purves, 1960; Oelschläger, 1986). Unfortunately, little has been published on either of these features in mysticetes, which makes it difficult to draw any firm conclusions.

Figure 11 Morphological features distinguishing “Metopocetus” vandelli from M. durinasus and M. hunteri.

Photographs show USNM 8518 (M. durinasus) and MUHNAC A1 (“M.” vandelli). Crania in dorsal view. Photograph of M. durinasus by RE Fordyce.

Fraser & Purves (1960: plates 6 and 7) show the small posterior sinus of extant Caperea marginata and Balaenoptera acutorostrata as occupying only a fraction of what remains of the paroccipital concavity in these taxa. If correct, then this would imply that the sinus cannot by itself account for the development of the paroccipital concavity as a whole. However, it needs to be noted that their assessment was largely based on the interpretation of osteological correlates and a previous description of B. acutorostrata (without any figures providing a detailed view of the posterior sinus) by Beauregard (1894), and hence may not be completely accurate. The ligamentous attachment of the stylohyal to the exoccipital in cetaceans has long been noted (Flower, 1885), and an enlargement of this structure seems particularly plausible in the case of Metopocetus with its well-developed tympanohyal. Nevertheless, it remains questionable whether the ligament would have filled the entire space defined by the paroccipital concavity. Additional data on the anatomy of this region in extant cetaceans are needed to determine what usually fills the paroccipital concavity in mysticetes, and thus ultimately what may have triggered it to grow so large in Metopocetus.

Figure 12 Basicranium of Eschrichtius robustus.

Left portion of the basicranium of the extant grey whale Eschrichtius robustus (USNM 364973) in ventrolateral view, highlighting the position of the paroccipital concavity.

Primary dorsal infraorbital foramen

All cetotheriids except neobalaenines and, perhaps, Cephalotropis, share the presence of an often enlarged, primary dorsal infraorbital foramen situated close to the base of the ascending process of the maxilla (e.g., Bouetel & De Muizon, 2006; El Adli, Deméré & Boessenecker, 2014; Gol’din, Startsev & Krakhmalnaya, 2014; Kimura & Hasegawa, 2010) (Fig. 13). In some taxa, such as Herpetocetus, and, possibly, Herentalia, a secondary foramen may also be present (Bisconti, 2015; Boessenecker, 2013; El Adli, Deméré & Boessenecker, 2014). Posteriorly, the foramen (or foramina) opens into a sulcus of variable length, which generally runs dorsally along the ascending process of the maxilla towards the cranial vertex. While the sulcus itself may be relatively short, the ascending process of the maxilla itself is often transversely concave (e.g., in Herentalia, Metopocetus, Piscobalaena, and to some degree also Herpetocetus), suggesting that the primary dorsal infraorbital foramen may supply a larger structure ascending along the maxilla towards the top of the cranium.

Figure 13 Primary dorsal infraorbital foramen of various cetotheriids.

Vertex of the cetotheriids Piscobalaena nana (MNHN SAS1616), Herpetocetus morrowi (UCMP 124950) and “Metopocetus” vandelli (MUHNAC A1) in dorsal view, showing the size and location of the primary dorsal infraorbital foramen.

Many mysticetes besides cetotheriids, including extant balaenids and balaenopterids, also possess what appears to be the homologue (or homologues) of the primary dorsal infraorbital foramen of cetotheriids; however, in these taxa the development of the foramen is often not as pronounced, often not accompanied by distinct sulci, and not as consistent (e.g., the foramen appears to be variable in Balaena mysticetus Linnaeus, 1758 and completely absent in Balaenella brachyrhynus Bisconti, 2005). The function of the primary dorsal infraorbital foramen is not entirely clear, especially in light of the fact that, at least in cetotheriids, it opens posterior to the level of the anterior border of the nasals (Fig. 13), and thus presumably cannot supply the nasal apparatus. Given the size of the foramen, as well as its consistent occurrence and the size and direction of the associated sulci (e.g., in Piscobalaena nana), it is tempting to speculate that the distinctive pattern of cetotheriid facial telescoping (i.e., posteriorly convergent maxillae resulting in shortened premaxillae and transversely compressed nasals) may at least partially have been driven by whatever soft tissue structure the foramen correlates with. Additional data on the function of the primary dorsal infraorbital foramen in living species may help to test this hypothesis.

Supplemental Information

Table S1 Palynomorph occurrences

Occurrences of marine and terrestrial palynomorphs in the matrix sample taken from the skull.

Click here for additional data file.

Supplemental Information 1 Complete molecular + morphological matrix

Click here for additional data file.

We thank Klaas Post for donating the specimen and allowing us to study it; Robert W. Boessenecker, Mark D. Uhen, Thomas A. Deméré and the editor Nicholas Pyenson for their helpful and constructive comments on the manuscript; R. Ewan Fordyce for insightful discussions and providing the photograph of Metopocetus durinasus; Seraina Klopfstein for technical advice on the total evidence analysis; Carl Buell for providing illustrative drawings of living and fossil cetaceans; and the staff of all of the institutions involved for access to material and help during our visits.

Institutional Abbreviations

MNHN Muséum national d’Histoire naturelle, Paris, France

MUHNAC Museu Nacional de História Natural e da Ciência, Lisbon, Portugal

NMR Natuurhistorisch Museum Rotterdam, the Netherlands

OU Geology Museum, University of Otago, Dunedin, New Zealand

UCMP University of California Museum of Paleontology, Berkeley, USA

USNM National Museum of Natural History, Smithsonian Institution, Washington, District of Columbia, USA

ZMT Fossil mammals catalogue, Canterbury Museum, Christchurch, New Zealand.

Additional Information and Declarations

Competing Interests

Author Contributions

Data Availability

New Species Registration

The authors declare there are no competing interests.

Felix Georg Marx and Stephen Louwye conceived and designed the experiments, performed the experiments, analyzed the data, contributed reagents/materials/analysis tools, wrote the paper, prepared figures and/or tables, reviewed drafts of the paper.

Mark E.J. Bosselaers conceived and designed the experiments, contributed reagents/materials/analysis tools, wrote the paper, prepared figures and/or tables, reviewed drafts of the paper, prepared the specimen.

The following information was supplied regarding data availability:

Illustrated cladistic scorings for the new material are available from MorphoBank, project 2225.

The following information was supplied regarding the registration of a newly described species:

publication: urn:lsid:zoobank.org:pub:E728C3DD-EB85-482F-ACE6-6558E3ED5441

species: urn:lsid:zoobank.org:act:391CF6D9-138C-4F88-AC4B-9903DA433FDA.

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
