# Peer review of "A new species of Metopocetus (Cetacea, Mysticeti, Cetotheriidae) from the Late Miocene of the Netherlands"

_PeerJ, doi:10.7717/peerj.1572_

## Round 0.1 · original submission · Minor Revisions

· Academic Editor

Minor Revisions

This manuscript has received three reviews, which found it meritorious of publication, but requiring fairly extensive revisions. On the whole, these concerns are all fairly minor, but their full implications may require revisiting some of the conclusions of this paper. Fortunately, none of the recommended changes are especially onerous, nor wholesale structural, but rather a consequence of specific changes requiring adjustment or clarification about details and/or the manuscript's scope. The preponderance of required changes thus tips towards a minor revision, which will require additional reviews, and potentially from different reviewers.

Overall, all of the reviewers caught some important details that need attention; in your revision, please explicitly address each one of their concerns, and if you and your co-authors disagree, please explain why. For example, Reviewer 2 requested important details about the locality and systematics that were not addressed; most critically, please explain why “Metopocetus” vandelli was not included in the phylogenetic analysis. Reviewer 1 rightly asked for more concrete judgement about the stratigraphic origin of Metopocetus durinasus, while Reviewer 3 asked for more information about comparisons to "Cetotherium" megalophysum, full disclosure of the character-taxon matrix (consider uploading this file to another digital repository, such as Data Dryad, figshare, or Zenodo) for subsequent revisions, and a more careful reading of El Adli & Deméré 2015 (Anat. Rec.), regarding the proposed insertion of the digastric muscle, which has an important role to play in proposed functional morphology of cetotheriid feeding. Most critically, I agree with Reviewer 3 that the description requires a bit more comparative nuance beyond qualitative trait assessments -- the addition of measurements for the arrangement and articulation of different bones would enhance the quality of this manuscript, as would more quantitative precision about interspecific differences from congeneric taxa.

I have also provided additional comments, mostly non-overlapping, in the attached PDF below, including the need for authority references, and stylistic suggestions to diction. Suggest also following the Society of Marine Mammalogy guidelines for common name usage (e.g., gray whale, instead of the British grey whale). Additionally:

Figure 1: As with the locality description (see Reviewer 2), this figure needs to be revised to include lat/lon coordinates placing this geography in coordinate space. The callout from Liessel should probably be in a box, with a panel label. Also, consider providing a bit more specificity about the geographic placement of the Netherlands -- don't assume that knowledge on the part of the reader. Cardinal directions should be included, along with appropriate scale bars, and reference to any grid/geodetic system if relevant.

Figures 2-9: Consider providing orthogonal anatomical orientation cues, especially for some of the more fragmentary remains that are challenging for even the specialist to correctly orient.

Figure 10. Please track for abbreviations from the figure, even if minor and obvious, in the caption (e.g., Plis = Pleistocene).

Figure 11. The specimen number is fairly obvious in the image, and therefore it would be appropriate to identify it as such in the caption (USNM 364973).

·

Basic reporting

So far as I can tell, the manuscrupt adheres to PeerJ standards and policies. The figures are all of high quality.

Experimental design

No ethical issues are evident.

Validity of the findings

From what I've read, all findings and speculation in the article are well-founded and reasoned.

Additional comments

This article is (unsurprisingly) a solid contribution to paleocetology, and I recommend publication with minor revisions. It is well written with little room for suggestions, but I have listed below an exhaustive list. Metopocetus has been a bit of a problematic baleen whale for a long time, and this new find from the Netherlands clarifies aspects of the incompletely known anatomy of M. durinasus. I'd like to offer the author sincere congratulations - well done!

However, the authors do not go so far as to apply their new fossil to solving the riddle of where in time Metopocetus durinasus fits - a fossil found in a riverbed in Maryland. I can't remember exactly who said what, but some authors suggested the specimen originated from the Calvert Fm. (Langhian), or alternatively the St. Mary's Fm. (Tortonian). The authors need to weigh in on this, as it is a bit of a nagging worry for paleocetologists, and the authors could go so far as to suggest that the late Miocene age of Metopocetus hunteri could support reinterpretation of M. durinasus as deriving from the St. Mary's Formation. This is perhaps my most major comment. The authors do not need to adopt this line of reasoning, but should at least discuss it in some capacity.

Another somewhat less major comment: the authors should assess, to the best of their abilities given incompleteness of the fossil, the ontogenetic age of the holotype since they are proposing a new species. The discussion need not be drawn out but could include observations on cranial suture closure and bone surface texture, development of muscle attachment crests, etc.

Minor comments:
77: The authors use "main ridge" and "involucral ridge" and thus should probably cite Oishi and Hasegawa 1995 for terminology here.
177-179: who collected the fossil, and when? Who prepared the fossil? All of these need to be addressed.
241: According to O. Lambert (can't remember paper) the sternmastoid fossa as traditionally identified in cetaceans actually houses the insertion for the sternocephalicus; if interested, R. E. Fordyce could be contacted for more information.
264: Is there any shallow pitting adjacent to the ext. occ. crest towards the vertex?
336: smooth involucra are typical of juvenile cetaceans, and whether or not the type is a juvenile should be addressed or referred to here if another cause for a smooth involucrum is more likely
346: change "abroad" into "a broad"
380: this reasoning does not follow exactly as written. The authors say (paraphrasing) "the analysis clearly places M. hunteri in the same genus as M. durinasus". Not to split hairs, but technically the analysis only means they are sister taxa and a less scrupulous taxonomist might still name a new genus for M. hunteri; perhaps some minor emendation along with a comment along the lines of "the two species are more parsimoniously interpreted as a single genus owing to their morphological similarity" would make this statement more correct.
407: could the expansion be quantified? "extremely" as a term should probably be eliminated from the MS as the word doesn't add anything - i.e. without quantifying it, "extremely expanded" can sort of mean the same thing as "expanded".
400-419: this discussion treats the posterior process simply as expanded or not, and might miss some important detail about the nature of the expansion. When considered simply as the posterior process widely exposed on the lateral wall of the skull, all Cetotheriidae ss. have this important synapomorphy - but the shape in lateral view differs. In herpetocetines, it's often circular or oval. In Cetotherium, Kurdalogonus, Brandtocetus, and Vampalus, it's nearly triangular or trapezoidal with a "spike" or wedge of bone pointing anterodorsally or posterodorsally - and in my (very recently published) paper on the eomysticetid Tokarahia (Boessenecker and Fordyce, 2015: Zoological Journal LS), even split out this character in my analysis to discriminate between the shape of posterior process exposure in different cetotheriids.
471: Boessenecker and Fordyce has now been published (2015)
503: add "to" after "owing"
505: El Adli et al. 2014 also inferred the length of baleen from the depth of the vomerine keel
Figure 1: is it possible to add a stratigraphic column?
Figure 10: there is a space missing between "occipital" and "crest"

Kind regards, Robert W. Boessenecker

·

Basic reporting

Good. There are a few comments in the manuscript to be addressed. I found the placement of the Biostratigraphy and Environment section to be oddly placed, and there is no real description of the rocks where the fossil was discovered. A bit more lithological context would be good.

Experimental design

Very good. I am not sure why "M." vandelli was not included in the phylogenetic analysis. The authors seem to know a lot about it, and comment on its phylogenetic placement, so I can't quite understand why it is missing.

Validity of the findings

These authors are using excellent data to reach new conclusions based on this new fossil that has been discovered.

Additional comments

Please see additional comments in the text. Also, please proofread the bibliography carefully. I found a few errors with just a brief reading.

·

Basic reporting

Overall, the manuscript is clearly written and appears to follow PeerJ guidelines. There are a few cases of awkward or colloquial wording (e.g., Lines 417 & 419). The photo-figures are very good, relevant, and well labeled, although Figure 8 appears to have been added after the in-text figure citations had been made (i.e., there are 12 actual figures, but in-text citations to only 11; this is also reflected in the figure captions). In two cases morpholoigcal features mentioned in the text are no labeled on the photo-figure (e.g., supramastoid crest, fig. 2B; and falciform process, fig. 5). The manuscript is not entirely self-contained in regards to the hypothesis that "Metopocetus" vandelli is closely related to "Cetotherium" megalophysum. Although the hypothesis is probably correct, there is insufficient morphological evidence presented in the limited discussion section (Lines 448-461) to support it. Without additional evidence and illustrations, this section should be removed. The "raw data" for the phylogenetic analysis part of the study is not readily available since the referenced MorphoBank file only contains character coding for the new taxon and none of the other taxa apparently included in the analysis. There is also a very limited discussion of the results of the phylogenetic analysis.

Experimental design

The methodology followed in the reported study is standard and clearly described.

Validity of the findings

There are some inconsistencies regarding anatomical nomenclature (e.g., skull vs. cranium vs. rostrum). The manuscript states that the specimen lacks the rostrum, however, it preserves the posterior portions of the maxilla (a part of the rostrum). The manuscript mentions the supraoribital process, but does not mention that this is part of the frontal. There is also a paucity of measurements of important anatomical features, which instead are merely described as large, small, deep, or thin. Actual measurements should be provided. There is also no description of the overall shape of the nasals. Overall, the Description section is good, although in only a few cases are direct comparisons made with other taxa. Typically, such comparisons might appear in the Discussion section, but for the most part such comparisons are lacking. This is especially a problem for the Phylogeny section of the Discussion, where detailed comparisons should be used as evidence in support of the authors' hypotheses of evolutionary relationships. The section on the Paroccipital concavity is speculative and includes an error concerning the origin of the digastric muscle in living gray whales. The origin of this muscle is actually lateral to the paroccipital concavity and not coincident with it. This factor challenges the hypothesis presented by the authors..

---

## Round 0.2 · Minor Revisions

· Academic Editor

Minor Revisions

This manuscript has now been reviewed by two previous reviewers, and the manuscript is now very close to acceptance, requiring just minor changes that should be very quick and will likely not be sent out to review again. Reviewer 1 asks to consider the use of a stratigraphic column, at the authors' discretion -- as this is a useful discussion (should the authors opt for public peer review), I encourage the authors to explain their decision on this matter. Reviewer 2 recommends two changes, one in the Systematic palaeontology section and the other a minor typo. For the former change, I disagree with Reviewer 2 because the concern arises out a misplaced understanding of crown group definitions for cetaceans: Neoceti is effectively crown Cetacea; Chaeomysticeti is more inclusive than crown Mysticeti, and thus the hierarchical structure presented in v1 is correct. In light of this confusion, I recommend adding an authority reference to the specific taxonomic concept for Cetacea, Chaeomysticeti, and Mysticeti entries, as done with Cetotheriidae (e.g., Mysticeti Gray, 1864 sensu...). The manuscript is otherwise in terrific shape and it is a thorough description that fully conforms with the publication criteria in PeerJ. I do strongly encourage the authors to consider making the review history public, as the authors' reply to the initial submission is valuable to specialists in its own right.

·

Basic reporting

The authors have done a great job and improved an already great manuscript into an excellent one that I recommend for acceptance into PeerJ. I do not need to see this manuscript again. There is only one last comment I have and it pertains to the use of a stratigraphic column. In response to my query whether a stratigraphic column could be added, the authors indicated that "the Breda Formation is a widespread, thick unit encompassing nearly all of the Dutch Miocene succession. In this particular case, a stratigraphic column showing the broader geological relationships of the type horizon would thus be somewhat uninformative."

From my perspective, the utilty of a stratigraphic column in vertebrate paleontology is different from its utility towards regional lithostratigraphy. The thickness of a mappable formation is irrelevant and reflects the relationship between basin subsidence rate and the persistence (or ephemerality) of a particular depositional setting - considerations that are tangential at best to vertebrate paleontology. Instead, a more practical application is succinctly illustrating vertical changes in depositional character as well as the relative vertical position of relevant paleoecological (e.g. depth- or temperature-informative invertebrates or microfossils) and age data. With the latter in mind, the reasons highlighted by the authors are a bit irrelevant. If useful paleoecological and age data exist above or below the type specimen but within the Breda Formation, a stratigraphic column is still useful. However, if all of the age and paleoecological data arise from the same exact horizon/locality as the Metopocetus hunteri holotype (as seems the case given the information presented in the manuscript), then I would agree that the inclusion of a stratigraphic column would be uninformative. I encourage the authors to consider whether other information might be useful to include in an additional figure - but I leave this entirely up to the author's discretion.

Well done! I look forward to seeing this published.

Kind regards,

Robert W. Boessenecker
College of Charleston, SC, USA

Experimental design

See above.

Validity of the findings

See above.

Additional comments

See above.

·

Basic reporting

No new comments.

Experimental design

No new comments.

Validity of the findings

No new comments.

Additional comments

I just made 2 comments in the manuscript. One is regarding the Systematic Paleontology, which is still incomplete, and now wrong. You accidentally placed Chaeomysticeti above rather than below Mysticeti, and I encourage you to add Neoceti as well. The other is a simple typo.

---

## Round 0.3 · Minor Revisions

· Academic Editor

Minor Revisions

I thank the authors for their careful reply to the last set of reviewer concerns. On the issue of stratigraphic context, I think that the authors provided a satisfying rejoinder to the reviewer's comments, and the addition to Figure 1 provides some much needed context. I would ask the authors to be a bit more precise in panel B and/or in the figure caption regarding the specifics about the stratigraphic origin of the type specimen (i.e., in addition to what is stated in the text). On the issue of taxonomic hierarchy, the situation has not improved. While the current hierarchy is not technically verboten, it is non-sensical, and only makes sense as an alternating blend of stem- and node-based taxonomic concepts, which is confusing and not immediately apparent to the non-specialist reader. I insist that the authors either change the hierarchy to make the nesting progressively more specific _clade_ names or to provide explicit author senses for each one. In your next revisions, please also de-highlight text from Table 1. I do not expect any further outside review will be necessary with the next revision.

---

## Round 0.4 · accepted · Accept

· Academic Editor

Accept

The corrections are acceptable, as is the manuscript. Again, I encourage the authors to consider making the review history to this manuscript publicly available, as I think there are some edifying discussions about stratigraphic precision in reporting fossil vertebrates.